# SS3DM: Benchmarking Street-View Surface Reconstruction with a Synthetic 3D Mesh Dataset

Yubin Hu[1][*]  Kairui Wen[1][*]  Heng Zhou[1]  Xiaoyang Guo[2]  Yong-Jin Liu[1][✉]

[1]Department of Computer Science and Technology, Tsinghua University
[2]Horizon Robotics

## Abstract

Reconstructing accurate 3D surfaces for street-view scenarios is crucial for applications such as digital entertainment and autonomous driving simulation. However, existing street-view datasets, including KITTI, Waymo, and nuScenes, only offer noisy LiDAR points as ground-truth data for geometric evaluation of reconstructed surfaces. These geometric ground-truths often lack the necessary precision to evaluate surface positions and do not provide data for assessing surface normals. To overcome these challenges, we introduce the SS3DM dataset, comprising precise **S**ynthetic **S**treet-view **3D M**esh models exported from the CARLA simulator. These mesh models facilitate accurate position evaluation and include normal vectors for evaluating surface normal. To simulate the input data in realistic driving scenarios for 3D reconstruction, we virtually drive a vehicle equipped with six RGB cameras and five LiDAR sensors in diverse outdoor scenes. Leveraging this dataset, we establish a benchmark for state-of-the-art surface reconstruction methods, providing a comprehensive evaluation of the associated challenges. For more information, visit our homepage at `https://ss3dm.top`.

## 1 Introduction

Reconstructing city-scale 3D meshes from street-view inputs is a challenging task in computer vision and graphics. While recent methods based on 3D Gaussians [23, 56] and NeRFs [29, 54] offer implicit proxies for novel view rendering, explicit mesh models remain indispensable for various industrial applications, including mixed reality, robotics, and gaming. Furthermore, the increasing use of closed-loop sensor simulations in autonomous driving scenarios [57, 52] has intensified the demand for high-precision city-scale mesh reconstructions.

To analyze the challenges in street-view surface reconstruction and enhance existing algorithms, it is crucial to benchmark these techniques using datasets that provide precise ground-truth mesh models. However, recent evaluations [37, 16] heavily rely on sparse LiDAR points from publicly available street-view datasets such as KITTI [10], Waymo [45], and nuScenes [5]. These evaluations encounter two main limitations. Firstly, the presence of random floaters and irregularities in the LiDAR points, caused by LiDAR sensor noise, hampers accurate geometric assessment. Secondly, the absence of surface normal information in LiDAR points poses challenges in evaluating the quality of reconstructed mesh models, since meshes with poor surface normal quality could appear geometrically invalid or ill-shaped, despite exhibiting good point-wise distance accuracy.

To mitigate these limitations, we propose SS3DM, a synthetic dataset specifically tailored for surface reconstruction of street-view outdoor scenes. SS3DM comprises meticulous ground-truth meshes of streets, buildings, and objects, facilitating the evaluation of surface reconstruction outcomes.

---

[*]These authors contributed equally to this work
[✉]Corresponding author.

38th Conference on Neural Information Processing Systems (NeurIPS 2024) Track on Datasets and Benchmarks.

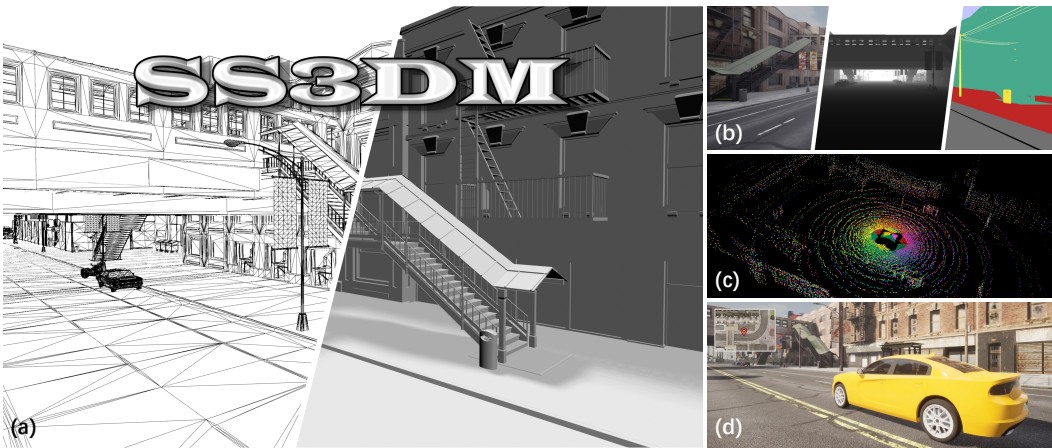

Figure 1: Overview of SS3DM: A 3D mesh dataset for benchmarking surface reconstruction of street-view outdoor scenes. a) High-fidelity 3D mesh models are provided for accurate geometric evaluation. b) SS3DM contains multi-view RGB video sequences which can be used as inputs for 3D surface reconstruction, along with depth and semantic information. c) Multi-view LiDAR points are also included as auxiliary inputs for 3D reconstruction. d) The street-view sequences are collected from the CARLA simulator with on-car sensors.

In real-world outdoor scenarios, obtaining accurate meshes for complex street-view structures is extremely challenging. In SS3DM, we address this challenge by developing a plugin that enables the direct export of detailed and precise 3D meshes from eight scenes in the CARLA simulator, an open-source driving simulator under MIT license. As illustrated in Figure 2, these precise 3D meshes exhibit finer structures compared to the LiDAR points provided in existing street-view datasets. This facilitates precise quantitative assessments of surface reconstruction methods.

The input data for street-view surface reconstruction in SS3DM consists of multi-view RGB and LiDAR sequences obtained from a virtual car in the CARLA simulator. The sensor specifications are simulated to align with advanced autonomous driving (AD) systems, as we believe they are suitable for collecting input data for street-view reconstructions. Specifically, we equip the virtual car with six RGB cameras and five LiDAR sensors (refer to Section 3.1 for more details). The car follows carefully planned routes, capturing a total of 28 sequences of varying lengths in eight different towns. The scenes within the dataset exhibit a variety of structures such as buildings, pedestrian overpasses, yards, fences, and poles, accurately reflecting real-world outdoor environments.

Leveraging the input data and mesh ground-truths, we conduct an extensive benchmark of state-of-the-art surface reconstruction methods for street-view scenes. Our benchmark incorporates comprehensive geometric evaluation metrics, including F-scores, Chamfer Distance, and Normal Chamfer Distance. Based on the experimental results, we extensively discuss limitations of existing methods and analyze the distinct challenges associated with street-view surface reconstruction.

To sum up, our contributions are twofold: 1) We introduce SS3DM, a synthetic dataset specifically designed for street-view surface reconstruction, consisting of photo-realistic synthetic video sequences, multi-view LiDAR points, and detailed ground-truth 3D meshes. 2) We extensively benchmark and analyze state-of-the-art surface reconstruction methods for outdoor scenes using SS3DM, and point out several outstanding directions, which are useful for developing future researches.

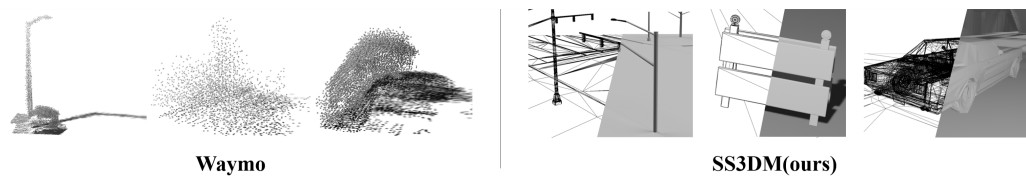

**Waymo**        **SS3DM(ours)**

Figure 2: Geometric ground-truths in Waymo (LiDAR points) and the proposed SS3DM (meshes).

## 2  Related Works

### 2.1  Street-View Datasets and Benchmarks

Several street-view datasets and benchmarks have been developed by researchers to address the challenges in autonomous driving (AD). Many of these datasets are primarily focused on visual perception tasks such as semantic segmentation and object detection [4, 6, 31, 63]. However, these datasets only provide video inputs and 2D annotations, which are not suitable for surface reconstruction benchmarks. In recent years, there has been an increasing popularity of multimodal datasets as most AD systems utilize various onboard sensors like LiDARs and IMUs. One of the most influential multimodal AD datasets is KITTI [10], which consists of 22 sequences of stereo videos and single-LiDAR points clouds. Other datasets have expanded the number of video cameras while still recording 3D environmental information with a single LiDAR [34, 5], or using two closely mounted LiDARs [20, 50]. To capture more geometric information, PandaSet [53] includes a front LiDAR in addition to the top LiDAR. A2D2 [11], Waymo [45], and ZOD [1] incorporate up to five LiDAR sensors positioned around the car. Although LiDAR sensors greatly enhance visual perception tasks like 3D object detection and point cloud segmentation, their LiDAR points are not accurate enough to be used as ground-truth for evaluating surface reconstruction due to sensor noise.

In the field of multi-view 3D reconstruction, there are several datasets and benchmarks available for large-scale scenes. However, most of these datasets are not specifically tailored for street-view surface reconstruction. Datasets such as Blended MVS [59], Mill 19 [47], UrbanScene3D [27], and OMMO [28] primarily consist of aerial sequences. On the other hand, datasets utilized by UrbanNeRF [37] are captured using human-carrying panorama cameras. The Waymo Block-NeRF [46] dataset includes street-view sequences captured by 12 realistic on-car cameras, but it lacks geometric ground-truth information. MatrixCity [25] provides ground-truth depth maps for synthetic street-view scenes, but the re-projected point clouds suffer from non-uniform distribution and lack accuracy, limiting their suitability for precise geometric evaluation.

SS3DM distinguishes itself by offering multi-RGB and multi-LiDAR street-view data, complemented by accurate ground-truth meshes. The inclusion of these features makes SS3DM particularly valuable for street-view surface reconstruction. For a comprehensive understanding of how SS3DM compares to existing datasets, please refer to Table 1.

### 2.2  Multi-View Surface Reconstruction Methods

The topic of surface reconstruction from multi-view images has been studied for decades. We briefly review the reconstruction methods and the underlying methodologies. Traditional surface reconstruction methods [3, 39] typically estimate the depth map of input images, and then perform point cloud fusion and surface reconstruction [22] as post-processing steps. Deep neural networks have enabled the prediction of depth maps from various sources, including monocular videos [68, 12, 13], multi-view images [21, 58, 19, 14], and multi-view video sequences [38]. Some recent works eliminate the intermediate point cloud representation and represent 3D surfaces as neural implicit SDFs (signed distance functions) [60, 49, 9, 26, 17, 61], optimizing them using neural rendering techniques inspired by NeRF [29]. Advanced novel view rendering methods like 3D Gaussian Splatting [23] have enabled the extraction of 3D surfaces from optimized 3D Gaussians in recent works [15, 18, 64], employing strategies such as marching tetrahedra [8, 42]. In addition

| Dataset | Source | Frames | Sequences | Avg. Duration | Cameras | LiDARs | GT Geometry |
|---|---|---|---|---|---|---|---|
| KITTI [10] | Real | 15k | 22 | **245s** | 4 | 1 | LiDAR Points |
| nuScenes [5] | Real | 40k | 1000 | 8s | 6 | 1 | LiDAR Points |
| PandaSet [53] | Real | 8.2k | 103 | 8s | 1 | 2 | LiDAR Points |
| Waymo [45] | Real | 200k | 1150 | 9s | 6 | **5** | LiDAR Points |
| ArgoVerse 2 [50] | Real | 150k | 1000 | 15s | **9** | 2 | LiDAR Points |
| ZOD [1] | Real | 100k | **1473** | 20s | 1 | 3 | LiDAR Points |
| MatrixCity [25] | Synthetic | **519k** | - | - | - | - | Depth Map |
| SS3DM | Synthetic | 81k | 28 | 48s | 6 | **5** | **Triangle Mesh** |

Table 1: Comparison between our SS3DM dataset with previous street-view datasets.

to image-only methods, certain approaches utilize auxiliary LiDAR points to regularize the implicit fields specifically for street-view scenarios, as seen in UrbanNeRF [37] and StreetSurf [16]. Other LiDAR-based mapping methods [41, 67, 48, 7] rely solely on sparse LiDAR supervision for street surface reconstruction. In our benchmark section, we evaluate several representative reconstruction methods using our SS3DM dataset, providing insights into their performance and suitability for street-view surface reconstruction.

## 3    SS3DM Dataset

The SS3DM dataset aims to introduce a new benchmark to the field of street-view surface reconstruction by providing accurate ground-truth mesh models along with input multi-view RGB and LiDAR sequences. Additionally, we also offer depth maps and semantic labels to support other tasks. In this section, we introduce the sensor specifications, the design of sequence trajectory, and the mesh exportation plugin in Section 3.1, 3.2 and 3.3, respectively.

### 3.1    Sensor Specifications

Within the CARLA simulator, we utilize a driving car as the agent to collect necessary input data for 3D surface reconstruction. The car is equipped with 6 RGB cameras and 5 LiDAR sensors, following the sensor settings employed in previous street-view datasets [5, 45]. Figure 3 illustrates the placement of the RGB cameras and LiDARs, which ensure a comprehensive coverage of the 360-degree surroundings while avoiding self-occlusion. Additionally, we export multi-view ground-truth depth maps and semantic labels by equipping specific sensors, which makes our dataset valuable for broader applications such as depth prediction [68, 13] and semantic segmentation [65, 35].

Table 2 presents the specifications of all the sensors used in SS3DM. The RGB images, depth maps, and semantic labels share the same camera parameters. For the LiDAR sensors, we incorporate noise simulation within CARLA and set the standard deviation to $\sigma_{noise} = 0.1$. These noisy LiDAR points can be utilized to evaluate the robustness of reconstruction methods with LiDAR inputs.

### 3.2    Data Collection

In our data collection process, we carefully plan the car trajectories to cover various scenes and buildings which are valuable and challenging for surface reconstructions. Within the 8 town scenes provided by CARLA, we capture 28 sequences of a wide range of environments, like city squares, large statues, and pedestrian bridges. To ensure the diversity of scene areas in SS3DM, we collected sequences of different lengths within each town. Our dataset consists of 14 short sequences with fewer than 300 frames, 8 medium length sequences ranging from 300 to 600 frames, and 6 long sequences consisting of 600 to 1000 frames. This diversity allows us to evaluate reconstruction methods for scenes at different scales. In total, SS3DM encompasses 13,535 data frames. Each data frame contains 6 RGB images, 5 LiDAR point cloud frames, 6 ground-truth depth images, and 6 ground-truth semantic segmentation maps. During data collection, the car autonomously navigates the pre-defined trajectories, capturing videos and LiDAR scans at 10 FPS. Figure 4 showcases the selected towns and the captured RGB images.

| Camera | Location | F, B | FL, FR | BL, BR |
|---|---|---|---|---|
| | Resolution | 1920×1080 | 1920×1080 | 1920×1080 |
| | FOV | 110° | 110° | 110° |
| | Camera Pitch | 0° | -15° | -15° |

| LiDAR | Location | T | F, B | L, R |
|---|---|---|---|---|
| | Channel | 32 | 32 | 32 |
| | Range | 100m | 50m | 50m |
| | Frequency | 10Hz | 10Hz | 10Hz |
| | Points / sec | 300k | 100k | 100k |
| | $\sigma_{noise}$ | 0.1 | 0.1 | 0.1 |

Table 2: Specifications for the on-car sensors.

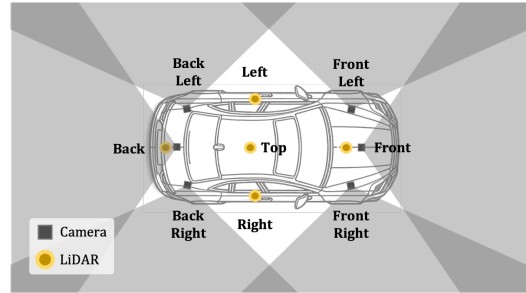

Figure 3: Camera and LiDAR locations.

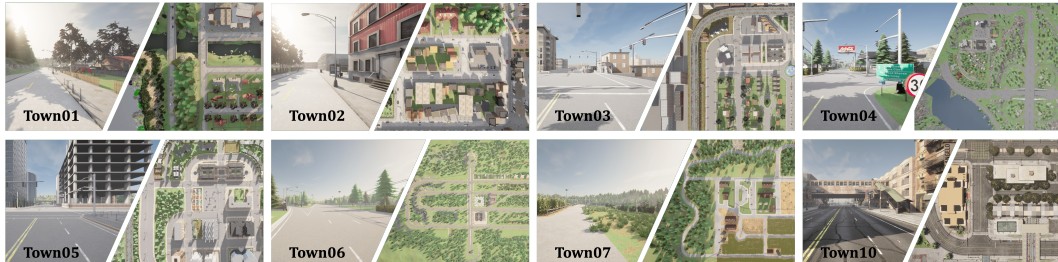

Figure 4: We collect our sequences in eight towns, including different types of areas and buildings. For each scene, we present a front camera image (left) and a bird's-eye view of the entire town (right).

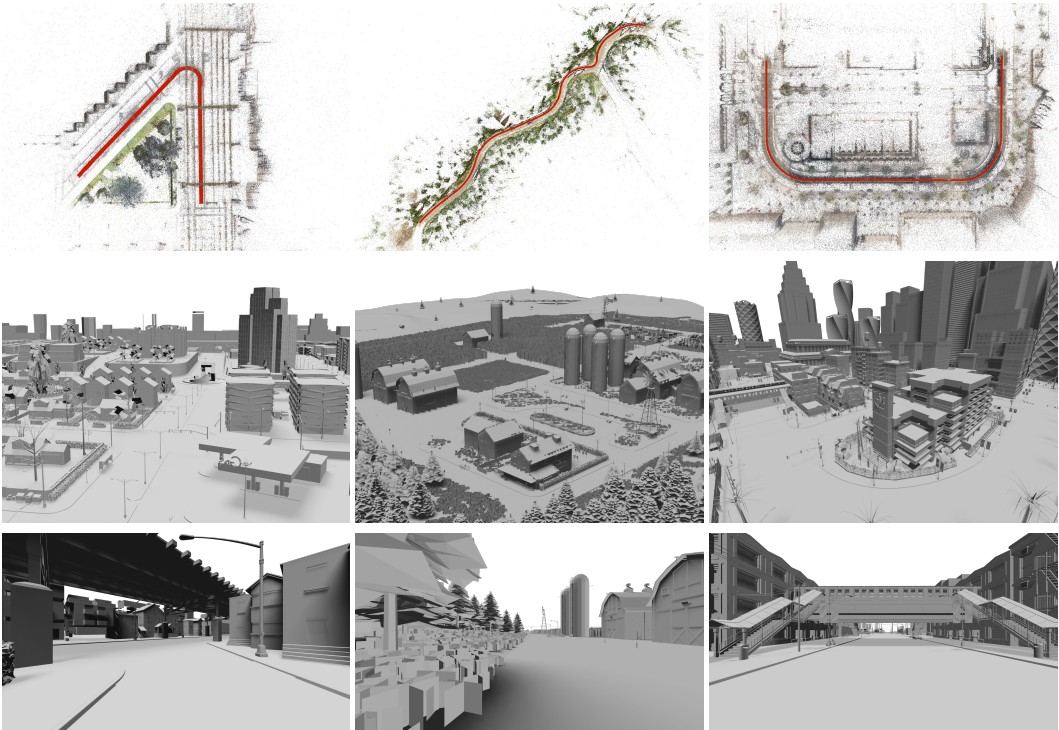

Figure 5: Visualization of the camera trajectories and ground-truth mesh models in SS3DM dataset. Top row: The camera trajectories and sparse point clouds reconstructed by Colmap from our ground-truth camera poses. Middle row: Global views of the ground-truth mesh models. Bottom row: On-the-ground views of the ground-truth mesh models.

We provide the intrinsic and extrinsic matrices for all cameras, as well as the rotation and translation matrices for all LiDAR sensors, serving as the ground-truth poses. These sensor poses in the OpenCV coordinate system can be directly utilized in downstream applications such as 3D reconstruction or employed to benchmark odometry algorithms [32, 66]. To validate the accuracy of our sensor poses, we conducted COLMAP sparse reconstruction [40] based on these cameras poses. The correct reconstruction results depicted in Figure 5 demonstrate the accuracy of the camera poses. We also verified the LiDAR poses by confirming the alignment between the transformed point clouds and the ground-truth mesh model. Further illustrations and details can be found in the Appendix.

### 3.3 Exporting Ground-truth Mesh Models

A distinctive and crucial aspect of SS3DM is the inclusion of high-precision ground-truth 3D mesh models for the large-scale scenes. To achieve this, we developed a plugin that exports the mesh models from the CARLA Unreal Engine and aligns them with the coordinate system of the sensor

poses and LiDAR points. We are committed to publicly releasing this plugin and the entire data exportation pipeline, allowing for free usage. This contribution can be utilized to generate additional datasets for surface reconstruction purposes using the CARLA simulator and Unreal Engine.

Unlike the depth maps and LiDAR points, which have been traditionally regarded as ground-truth geometry in previous datasets, the exported triangle mesh models provide dense geometry representations of elements in the street-view scenes, including the flat road surfaces and intricate structures like light poles, parked cars, and bus stations. With the availability of these mesh models, we can uniformly sample point clouds of arbitrary density and calculate accurate surface normal vectors for each point, which could be utilized in position and surface normal evaluations.

## 4 Experiments

In this section, we present our benchmark on street-view surface reconstruction based on the proposed SS3DM dataset. Specifically, we utilize all data sequences in SS3DM as the test data. By benchmarking state-of-the-art methods on sequences of varying lengths, we uncover some key challenges for surface reconstruction for street-view surface reconstruction, and suggest potential directions for future research in this field.

### 4.1 Evaluated Methods

Our experiments primarily focus on evaluating state-of-the-art methods in the street-view surface reconstruction context. We select representative approaches from various methodologies, including R3D3 [38] for multi-view-stereo, NeRF-LOAM [7] for LiDAR-based mapping, UrbanNeRF [37] for NeRFs, StreetSurf [16] for NeuralSDFs, and SuGaR [15] for 3D Gaussians. We provide a detailed discussion of the reasons for selecting these methods and the technical details of them in the Appendix.

### 4.2 Evaluation Protocol

To evaluate the aforementioned approaches on our benchmark sequences, we input the data frames of all time steps into the algorithm pipeline without temporal re-sampling. For methods that rely solely on LiDAR inputs (NeRF-LOAM and StreetSurf (LiDAR)), we input the point clouds collected by 5 LiDAR sensors. For methods that rely solely on RGB images (R3D3, StreetSurf (RGB), and SuGaR), we only input the multi-camera video frames. The remaining methods, UrbanNeRF and StreetSurf (Full), take both modalities as input.

***Resampling.*** To uniformly assess the reconstruction accuracy of each triangle face, we densely sample point clouds from the ground-truth and reconstructed mesh surfaces, rather than sampling from the triangle vertices. Since the reconstruction methods are not expected to reconstruct occluded surfaces, we first filter out invisible triangle faces based on the camera poses before the resampling step. After filtering, we oversample 10.24 million points using a uniform strategy that approximately distributes the sampled points according to the area of each triangle face. The over-dense point clouds are then resampled using a voxel size of $\tau = 0.05m$ for precise evaluation.

***Cropping.*** Finally, we crop the point clouds using a 3D bounding box calculated by extending the bounding box of camera trajectories by 25m in each direction. This cropping strategy has two main reasons: 1) Current methods often struggle when reconstructing distant surfaces, so evaluating distant points becomes meaningless. 2) Too many distant points within the benchmarking point clouds can dominate the metric numbers due to their low performance, which obscures the significant performance gaps between methods when reconstructing nearby surfaces.

***Metrics.*** We employ a comprehensive set of metrics to assess the quality of the reconstructed surfaces, including Intersection over Union (IoU), F-score, Chamfer Distance (CD), Normal Chamfer Distance (CD$_N$), and their respective sub-terms. The IoU metrics are computed following the volumetric IoU in [43] with a voxel size of $0.10m$. F-score is defined as the harmonic mean of precision and recall following [24], where recall is the fraction of points on ground truth mesh surface that lie within a threshold distance to the predicted mesh surface, and precision is the fraction of points on predicted mesh that lie within a threshold distance to the ground truth mesh. Specifically, we set the threshold in F-score to $\tau = 0.05m$, which is the same as the voxel size used for resampling.

| | Method | IoU ↑ | Prec. ↑ | Recall ↑ | F-score ↑ | Acc ↓ | Comp ↓ | CD ↓ | Acc$_N$ ↓ | Comp$_N$ ↓ | CD$_N$ ↓ | CD + CD$_N$ ↓ |
|---|---|---|---|---|---|---|---|---|---|---|---|---|
| Short Seq. | R3D3 | 0.003 | 0.007 | 0.011 | 0.009 | 0.912 | 0.910 | 1.822 | 0.670 | 0.662 | 1.332 | 3.154 |
| | UrbanNeRF | 0.063 | 0.125 | 0.177 | 0.142 | 0.372 | 0.513 | 0.885 | 0.358 | 0.482 | 0.839 | 1.725 |
| | SuGaR | 0.052 | 0.105 | 0.091 | 0.093 | 0.361 | 0.380 | 0.741 | 0.578 | 0.607 | 1.185 | 1.926 |
| | StreetSurf (RGB) | 0.057 | 0.106 | 0.090 | 0.093 | 0.345 | 0.445 | 0.791 | 0.417 | 0.558 | 0.974 | 1.765 |
| | NeRF-LOAM | 0.094 | 0.147 | 0.184 | 0.157 | **0.139** | 0.367 | 0.507 | 0.642 | 0.694 | 1.336 | 1.843 |
| | StreetSurf (LiDAR) | 0.157 | **0.290** | **0.373** | **0.314** | 0.211 | 0.312 | 0.523 | 0.434 | 0.520 | 0.953 | 1.476 |
| | StreetSurf (Full) | **0.166** | 0.277 | 0.326 | 0.287 | 0.175 | **0.310** | **0.485** | **0.311** | **0.466** | **0.777** | **1.262** |
| Medium Seq. | R3D3 | 0.002 | 0.006 | 0.006 | 0.006 | 0.866 | 0.917 | 1.784 | 0.741 | 0.743 | 1.484 | 3.268 |
| | UrbanNeRF | 0.040 | 0.069 | 0.102 | 0.080 | 0.456 | 0.598 | 1.054 | 0.493 | 0.580 | 1.073 | 2.127 |
| | SuGaR | 0.018 | 0.043 | 0.022 | 0.028 | 0.470 | 0.508 | 0.978 | 0.697 | 0.694 | 1.391 | 2.369 |
| | StreetSurf (RGB) | 0.043 | 0.065 | 0.061 | 0.061 | 0.351 | 0.495 | 0.847 | 0.565 | 0.635 | 1.201 | 2.047 |
| | NeRF-LOAM | 0.062 | 0.082 | 0.120 | 0.093 | **0.158** | 0.397 | **0.555** | 0.707 | 0.742 | 1.449 | 2.004 |
| | StreetSurf (LiDAR) | 0.076 | **0.153** | **0.161** | **0.153** | 0.262 | 0.379 | 0.641 | 0.542 | 0.609 | 1.151 | 1.792 |
| | StreetSurf (Full) | **0.085** | 0.143 | 0.147 | 0.141 | 0.210 | 0.395 | 0.605 | **0.475** | **0.576** | **1.050** | **1.656** |
| Long Seq. | R3D3 | 0.001 | 0.004 | 0.003 | 0.003 | 0.910 | 0.970 | 1.880 | 0.793 | 0.788 | 1.581 | 3.461 |
| | UrbanNeRF | 0.012 | 0.018 | 0.025 | 0.021 | 0.540 | 0.687 | 1.228 | **0.572** | 0.701 | 1.273 | 2.501 |
| | SuGaR | 0.005 | 0.021 | 0.005 | 0.007 | 0.604 | 0.627 | 1.231 | 0.758 | 0.746 | 1.504 | 2.734 |
| | StreetSurf (RGB) | 0.016 | 0.031 | 0.019 | 0.023 | 0.460 | 0.588 | 1.047 | 0.686 | 0.727 | 1.412 | 2.460 |
| | NeRF-LOAM | 0.035 | 0.049 | 0.059 | 0.053 | **0.167** | 0.482 | **0.649** | 0.763 | 0.772 | 1.535 | 2.185 |
| | StreetSurf (LiDAR) | 0.033 | **0.082** | 0.059 | **0.068** | 0.308 | 0.478 | 0.786 | 0.626 | 0.691 | 1.317 | 2.103 |
| | StreetSurf (Full) | **0.040** | 0.077 | **0.061** | **0.068** | 0.253 | **0.465** | 0.718 | **0.572** | **0.670** | **1.242** | **1.960** |
| Mean | R3D3 | 0.003 | 0.006 | 0.008 | 0.007 | 0.898 | 0.925 | 1.823 | 0.717 | 0.712 | 1.429 | 3.252 |
| | UrbanNeRF | 0.046 | 0.086 | 0.123 | 0.098 | 0.432 | 0.575 | 1.007 | 0.442 | 0.557 | 0.999 | 2.006 |
| | SuGaR | 0.032 | 0.069 | 0.053 | 0.056 | 0.444 | 0.469 | 0.914 | 0.650 | 0.662 | 1.312 | 2.226 |
| | StreetSurf (RGB) | 0.044 | 0.078 | 0.067 | 0.069 | 0.372 | 0.490 | 0.862 | 0.517 | 0.616 | 1.133 | 1.995 |
| | NeRF-LOAM | 0.072 | 0.107 | 0.139 | 0.116 | **0.151** | 0.400 | **0.551** | 0.687 | 0.724 | 1.411 | 1.962 |
| | StreetSurf (LiDAR) | 0.107 | **0.206** | **0.245** | **0.215** | 0.246 | 0.367 | 0.613 | 0.506 | 0.582 | 1.088 | 1.701 |
| | StreetSurf (Full) | **0.116** | 0.196 | 0.218 | 0.198 | 0.202 | **0.367** | 0.569 | **0.414** | **0.541** | **0.955** | **1.524** |

Table 3: IoUs (0.10m), F-scores (0.05m), Chamfer Distances (m), and Normal Chamfer Distances for each method on the benchmark dataset. The sub-terms of each metric are also listed for detailed analysis, including Precision, Recall, Accuracy and Completeness. We find that the summation of Chamfer Distance and Normal Chamfer Distance can describe the reconstruction quality better, which is more close to the qualitative results presented in Figure 6. The evaluated methods are grouped according to their input data modalities: RGB, LiDAR, and RGB+LiDAR.

Chamfer Distance (CD) measures the average distance between two point sets in meters. Specifically, the CD between ground-truth point cloud $G$ and predicted point cloud $P$ is defined as

$$\text{CD} = \text{Acc} + \text{Comp} = \sum_{p \in P} \min_{g \in G} \text{D}_\text{E}(p, g) + \sum_{g \in G} \min_{p \in P} \text{D}_\text{E}(g, p), \tag{1}$$

where Acc and Comp denote the Accuracy and Completeness term of CD, respectively. $\text{D}_\text{E}(\cdot, \cdot)$ denotes the Euclidean Distance. To evaluate the surface normals, we derive the Normal Chamfer Distance between G and P by replacing the distance metric $\text{D}_\text{E}(\cdot, \cdot)$ in Equation 1 with the Cosine Distance between the surface normal vectors of the point pairs, which can be formulated as:

$$\begin{aligned} \text{CD}_N = \text{Acc}_N + \text{Comp}_N &= \sum_{p \in P} \text{D}_\text{C}(\mathbf{n}_p, \mathbf{n}_{g_p}) + \sum_{g \in G} \text{D}_\text{C}(\mathbf{n}_g, \mathbf{n}_{p_g}), \\ g_p = \arg\min_{g \in G} \text{D}_\text{E}(p, g), &\quad p_g = \arg\min_{p \in P} \text{D}_\text{E}(g, p), \end{aligned} \tag{2}$$

where $\mathbf{n}_a$ denotes the surface normal vector of point $a$, and $\text{D}_\text{C}(\mathbf{n}_a, \mathbf{n}_b) = 1 - (\mathbf{n}_a \cdot \mathbf{n}_b)/(||\mathbf{n}_a|| \cdot ||\mathbf{n}_b||)$ denotes the Cosine Distance.

### 4.3 Surface Reconstruction Results

***Quantitative Results.*** Table 3 provides a summary for the performance of the evaluated methods on different subsets of SS3DM with varying sequence lengths. Notably, all methods exhibit lower performance on longer sequences. On the subset of long sequences, none of the evaluated methods achieve an F-score higher than 0.1, indicating the current struggle in reconstructing 3D surfaces from long sequences. These findings demonstrate street-view surface reconstruction remains a highly challenging task. Our SS3DM dataset provides a valuable benchmark for future researchers to thoroughly evaluate their algorithms.

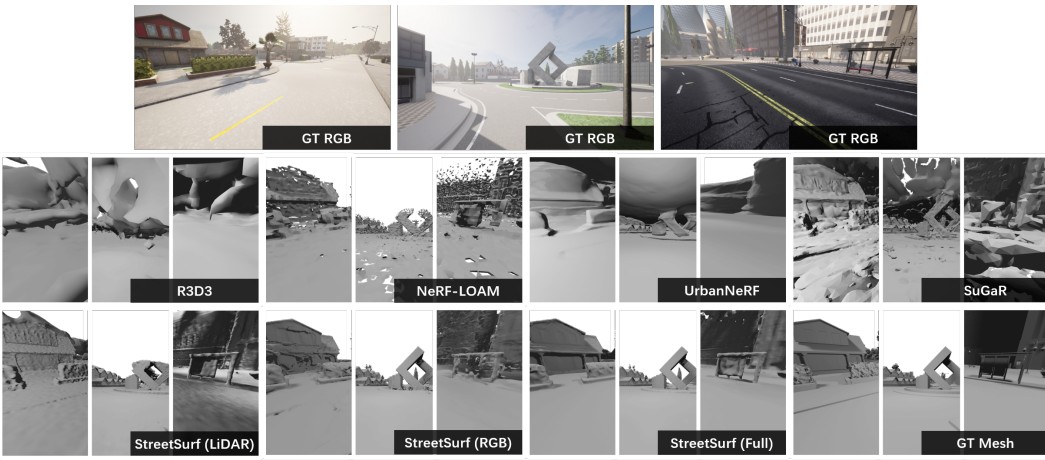

Figure 6: Qualitative comparisons for reconstruction results of evaluated methods on selected view in Town01_150 (left), Town03_360 (middle), and Town10_1000 (right).

***Qualitative Results.*** We showcase the reconstruction results rendered from specific camera viewpoints of RGB images in Figure 6 and visually depict the point clouds utilized for metric calculations in Figure 7. Notably, the presence of "floaters" in reconstructed surfaces significantly undermines the accuracy of the reconstruction results. Additionally, inaccuracies in reconstructing sparsely observed regions, such as the extremities of Town01_150 and the central areas of Town10_1000, have a substantial negative impact on the evaluation metrics. Looking ahead, integrating strategies to address "floaters" and enhance reconstructions in sparsely observed regions holds promise for improving the overall quality of street-view surface reconstructions.

When comparing the qualitative and quantitative results, we observe that the distance metrics, F-score and CD, cannot reflect the actual reconstruction quality reflected by the qualitative visualizations. On average, StreetSurf (LiDAR) and NeRF-LOAM achieve the highest F-score and lowest CD, respectively. But both methods produce worse reconstructed surfaces than StreetSurf (Full). On the contrary, we note that the surface normal metric $CD_N$ is more relevant to the actual reconstruction quality, which demonstrates the importance of surface normal evaluation based on our accurate ground-truth mesh models. To provide a comprehensive measurement of both distance metrics and surface normal metrics, we also report the summation of CD and $CD_N$ in Table 3, in terms of which StreetSurf (Full) also achieves the best average performance.

## 4.4 Discussion

Among these evaluated methods, R3D3 achieves lower reconstruction quality than other methods. The predicted per-frame depth maps are erroneous and result in noisy and inaccurate surfaces. As for the LiDAR-based mapping method NeRF-LOAM, it achieves excellent Accuracy metric and performs well in terms of overall Chamfer Distance. However, the reconstructed surfaces exhibit significant noise, as demonstrated by the high $CD_N$ scores. UrbanNeRF produces flat surfaces with good $CD_N$ results due to the inherent smoothness of MLP representations. However, the MLPs fail to capture precise structures, resulting in over-smoothed mesh models. While SuGaR demonstrates good results in object-level scenes, as reported in the original paper, it fails to reconstruct high-quality surfaces when applied to large-scale outdoor scenes. Figure 6 showcases the presence of bubble-like structures inherited from the 3D Gaussians in the reconstructed road ground. Additionally, the severe floaters in the sky negatively impact the evaluation metrics.

In comparison, StreetSurf (Full) represents the scene by NeuralSDF and enhances the representation ability by incorporating multi-level hash grid features. Consequently, this method reconstructs superior surfaces for both flat roads and intricate structures, achieving the best performance in terms of the CD + $CD_N$ metric. By removing the LiDAR and RGB inputs separately, we find that both modalities contribute to the final performance of StreetSurf (Full). Although the LiDAR-only variant achieves a better F-score, it falls short in average distance metrics. While StreetSurf (Full) achieves a high level of reconstruction quality, it fails to capture delicate structures that can be reconstructed

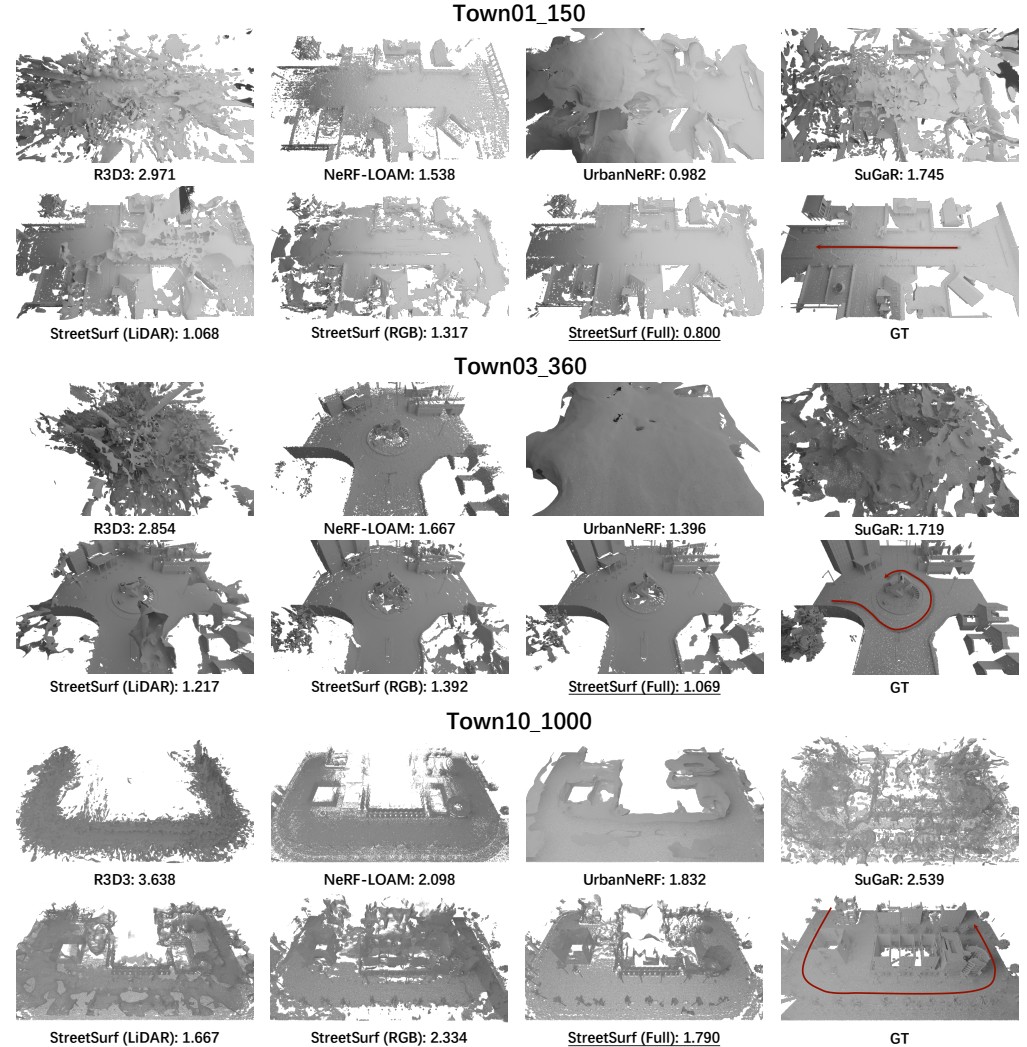

Figure 7: Comparison of the resampled and cropped point clouds for evaluation purposes. The CD + $CD_N$ metric is annotated next to the name of each method, with the method achieving the best reconstruction quality on the sequence underlined. Additionally, the vehicle trajectories are depicted as red arrows in the ground truth point clouds.

by the LiDAR-only and RGB-only counterparts, as illustrated in Figure 6. Therefore, it is crucial to explore better combinations of the RGB and LiDAR input modalities in future research. Furthermore, many other technical aspects employed in StreetSurf hold value for further advancements in surface reconstruction methods for large-scale scenes. For example, the planar SDF initialization helps eliminate floaters in the sky, and the supervision of monocular surface normal maps improves the smoothness of reconstructed surfaces.

## 4.5 Future Directions for Street-View Surface Reconstruction

Based on the benchmarking results, we list several research directions that we believe should be pursued in future methods for large-scale surface reconstruction.

*Efficient Representations:* Dense representations like voxel grids, as utilized in NeRF-LOAM, consume significant GPU memory for large-scale scenes. StreetSurf addresses this by employing hash feature grids proposed in Instant-NGP [30] to compactly encode the entire scene. However, hash features do not explicitly save redundant features allocated to empty spaces. Furthermore, an

efficient representation should allocate finer-grained features to delicate structures such as cars and light poles. Therefore, exploring adaptive and efficient representations for surface reconstruction is crucial. This could involve investigating sparse representations [44] or hierarchical structures [62] specifically tailored for large-scale surface reconstruction.

***Split-and-Merge Strategy:*** Another promising research direction is the exploration of split-and-merge strategies for large-scale surface reconstruction. Splitting the scene into smaller, manageable parts and then merging them back together can help alleviate the computational burden and memory demands associated with processing massive datasets. Drawing insights from previous methods designed for large-scale scene rendering [46, 47] could provide valuable guidance.

***Multi-stage Reconstruction:*** A third crucial research direction to explore is the development of multi-stage reconstruction methods. By decomposing the reconstruction process into multiple coarse-to-fine stages, the pipeline can focus on the smoothness and flatness of planar regions in the early stages. As the trianing progresses, more attention can be given to detailed objects, enabling precise reconstruction of their intricate details. This strategy allows for a good balance between achieving smoothness in planar areas and capturing rich details for intricate structures, resulting in greater accuracy and fidelity.

## 5   Conclusion

In this study, we have built SS3DM, a synthetic street-view dataset containing precise 3D ground-truth meshes that is specifically designed for evaluating surface reconstruction techniques in street-view outdoor scenes. The dataset comprises synthetic multi-camera videos, multi-view LiDAR points, and accurate 3D ground-truth meshes captured in a diverse range of outdoor environments. Leveraging SS3DM, we conducted a comprehensive benchmark of state-of-the-art surface reconstruction methods, revealing their limitations in terms of point-wise distance accuracy and surface normal accuracy. These findings provide insights into the challenges of large-scale outdoor modeling and potential directions for future research.

**Limitations.** Currently, the dataset has limited scene diversity and does not support dynamic object reconstruction, such as moving cars and pedestrians. To address these limitations, future enhancements are planned, including exporting per-frame 3D meshes for dynamic objects and incorporating additional scenes from different simulators.

## 6   Acknowledgements

This work was supported by the National Science and Technology Major Project (2022ZD0117904), and the Natural Science Foundation of China (Project Number U2336214).

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

# A  Appendix

## A.1  Essential Dataset Details

Our dataset, along with its Croissant metadata can be accessed here[1]. Currently it only contains part of the dataset, full dataset as well as the metadata will be released later. The dataset adopts the same format as StreetSurf. Details of the format can be found at this link[2].

Our dataset adheres to the CC BY 4.0 license. We will continue to update the dataset, incorporating such as dynamic objects, dynamic weather conditions, and more.

## A.2  Visualization of Aligned LiDAR Points

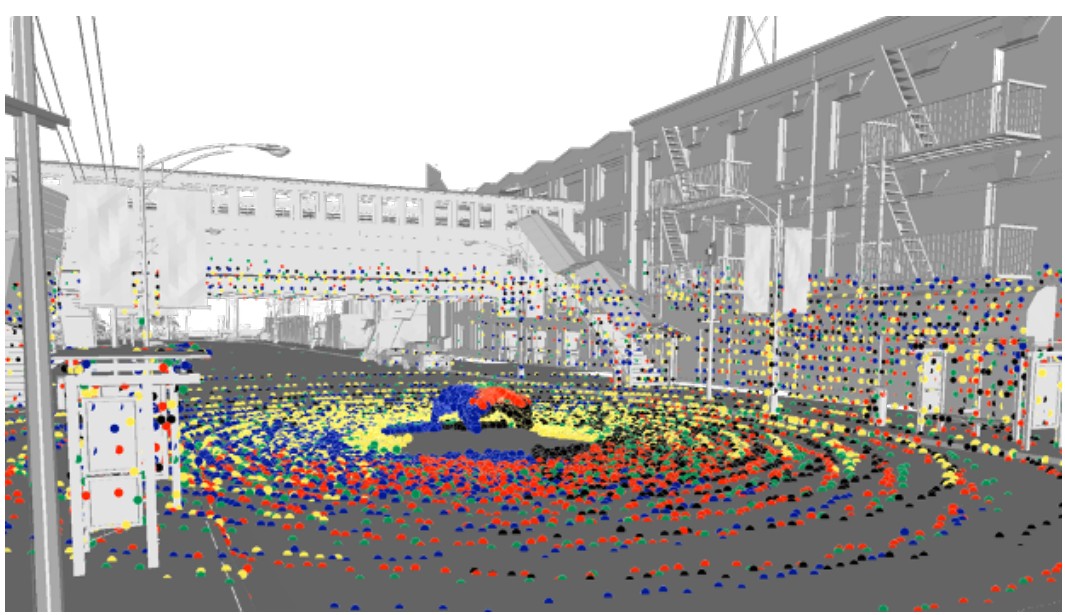

Figure 8: Visualization showcasing the alignment between the ground truth mesh model and point clouds obtained from multiple LiDAR sensors at a single timestep. The colors in the point clouds distinguish data collected from different LiDAR sensors: yellow represents Front, black represents Right, red represents Back, blue represents Left, and green represents Top.

To confirm the correctness of our exported translation and rotation matrices for ground truth LiDAR poses, we visualize the ground truth mesh model and multi-LiDAR point clouds in Figure 8. The good alignment between LiDAR points and mesh models verifies the accuracy of our exported LiDAR poses. These LiDAR point clouds and accurate LiDAR poses are also valuable for evaluating point cloud registration algorithms [2, 33, 36] in the street-view scenes.

## A.3  Evaluated Methods

We have selected representative approaches from various methodologies, including multi-view stereo, LiDAR-based mapping, NeRFs, NeuralSDFs, and the emerging 3D Gaussians. All trainings and evaluations were conducted with a single 80GB Nvidia A100 GPU.

R3D3 [38] is a recent method that performs dense 3D reconstruction and ego-motion estimation from multi-camera video sequences. In contrast to depth estimation methods based on monocular depth estimation [68, 12, 13] and multi-view stereo [39, 21, 58, 19, 14], R3D3 leverages correlation information in both spatial and temporal dimensions. In our experiments, we fix the ground-truth camera poses and predict depth maps using the officially provided checkpoint pre-trained on nuScenes

---

[1]`https://ss3dm.top`
[2]`https://github.com/AlbertHuyb/neuralsim/blob/main/docs/data/autonomous_driving.`
`md`

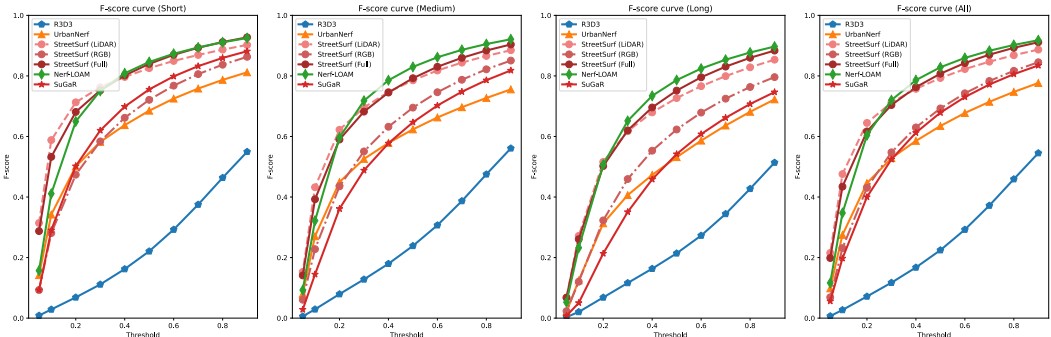

Figure 9: The F-score metrics across various thresholds, spanning from 0.05m to 0.9m.

[5]. Subsequently, we re-project and fuse the depth maps into point clouds, from which we extract mesh models for evaluation using Poisson Surface Reconstruction [22].

NeRF-LOAM [7] is a state-of-the-art method that employs neural implicit representation for LiDAR-based odometry and mapping. We select NeRF-LOAM as a representative of LiDAR-based mapping methods [41, 67, 48] and evaluate it with ground-truth camera poses, utilizing the publicly released code.

UrbanNeRF [37] models the geometry of large-scale scenes using the density field of NeRF represented by a single MLP. In comparison to other NeRF-based methods for large-scale scenes [46, 51, 55], UrbanNeRF incorporates geometric supervision from LiDAR point clouds to enhance surface reconstruction. We implement UrbanNeRF based on the implementation details provided in the original paper.

StreetSurf [16] applies NeuralSDF to model the implicit geometry and employs hash voxel features to enhance representation capability. In addition to RGB images and LiDAR points, StreetSurf utilizes monocular surface normal predictions as auxiliary supervision to improve the quality of reconstructed geometry. We evaluate three modes of StreetSurf using the publicly released code: LiDAR-only mode, RGB-only mode, and Full mode.

SuGaR [15] is a surface reconstruction method based on 3D Gaussian Splatting [23], which models the scene as 3D Gaussians and aligns the mesh model to the optimized Gaussian field. We evaluate this method to explore the potential of applying 3D Gaussians to surface reconstruction of large-scale scenes. In our experiments, we utilize the official code of SuGaR to optimize 3D Gaussians for 7k iterations and then perform the coarse-to-fine surface reconstruction pipeline. To meet the requirements for input format of SuGaR, we convert our data sequences to the Colmap format, which includes our ground truth camera poses and the sparse point clouds produced by Colmap sparse reconstruction for 3D Gaussian initialization.

### A.4 F-score Curves

To provide a comprehensive analysis of reconstruction accuracy, we evaluate the F-score metrics using ten different thresholds and present the corresponding curves in Figure 9. The evaluated thresholds include 0.05m, 0.1m, 0.2m, 0.3m, 0.4m, 0.5m, 0.6m, 0.7m, 0.8m, and 0.9m.

The F-score metrics, varying with different thresholds, offer insights into the performance of reconstruction methods from various perspectives. For instance, F-score (0.05m) measures the accuracy of reconstructed surfaces within a low tolerance for errors, as discussed in Section 4. Conversely, F-score (0.9m) reflects the presence of distant floaters in the reconstructed surfaces. Methods with lower F-score (0.9m) tend to reconstruct more floaters in the distant areas. As depicted in Figure 9, both R3D3 and UrbanNeRF exhibit lower F-score (0.9m) compared to other methods, indicating the presence of more floaters in their reconstructed surfaces, as depicted in Figure 6.

Regarding the overall tendencies across F-scores for all thresholds, we observe that StreetSurf (Full) outperforms other methods for small thresholds ranging from 0.05m to 0.2m but performs worse than NeRF-LOAM for larger thresholds. This phenomenon suggests that while StreetSurf (Full) reconstructs more accurate surfaces near the ground truth compared to the LiDAR-mapping method

NeRF-LOAM, it tends to generate more structures that deviate from the ground truth in distant regions. Future work could aim to combine the strengths of StreetSurf (Full) in nearby regions with those of NeRF-LOAM in distant regions to achieve improved results.

### A.5 More Visualizations of the Dataset

**Dynamic Objects.** We have started to extend SS3DM during the rebuttal period by including dynamic objects and traffics utilizing the CARLA traffic functionalities. We could add moving objects in the street scenes and extract the ground truth meshes for dynamic objects at every timestamp. Please refer to Figures 10 and 11 for more visualizations. With the dynamic masks depicted in Figure 10, researchers could evaluate the reconstruction algorithms with occlusions like cars and pedestrians. Moreover, evaluations of dynamic object reconstruction could be further conducted base on the ground truth object-wise meshes as shown in Figure 11.

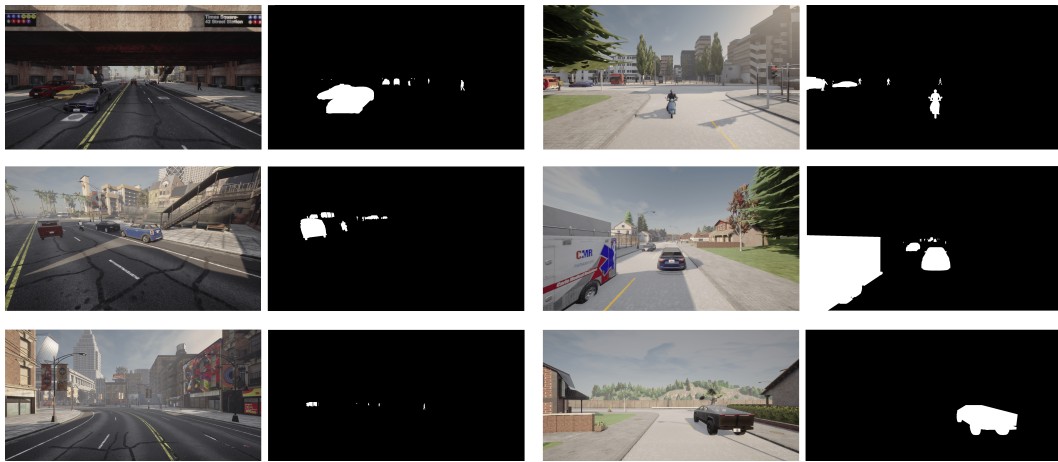

Figure 10: RGB images with dynamic objects and their respective dynamic masks.

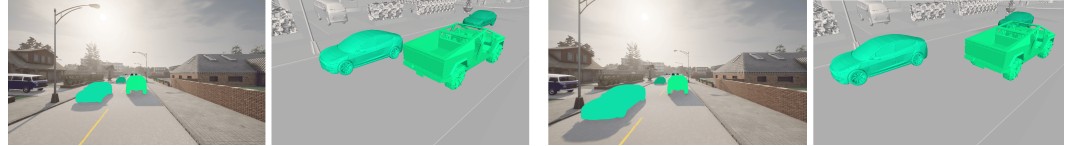

Figure 11: Visualization of ground truth dynamic object meshes at different timestamps.

**Fine-grained Structures.** We provide more visualizations of the complex and precise geometric structures included in the ground truth mesh of SS3DM in Figure 12.

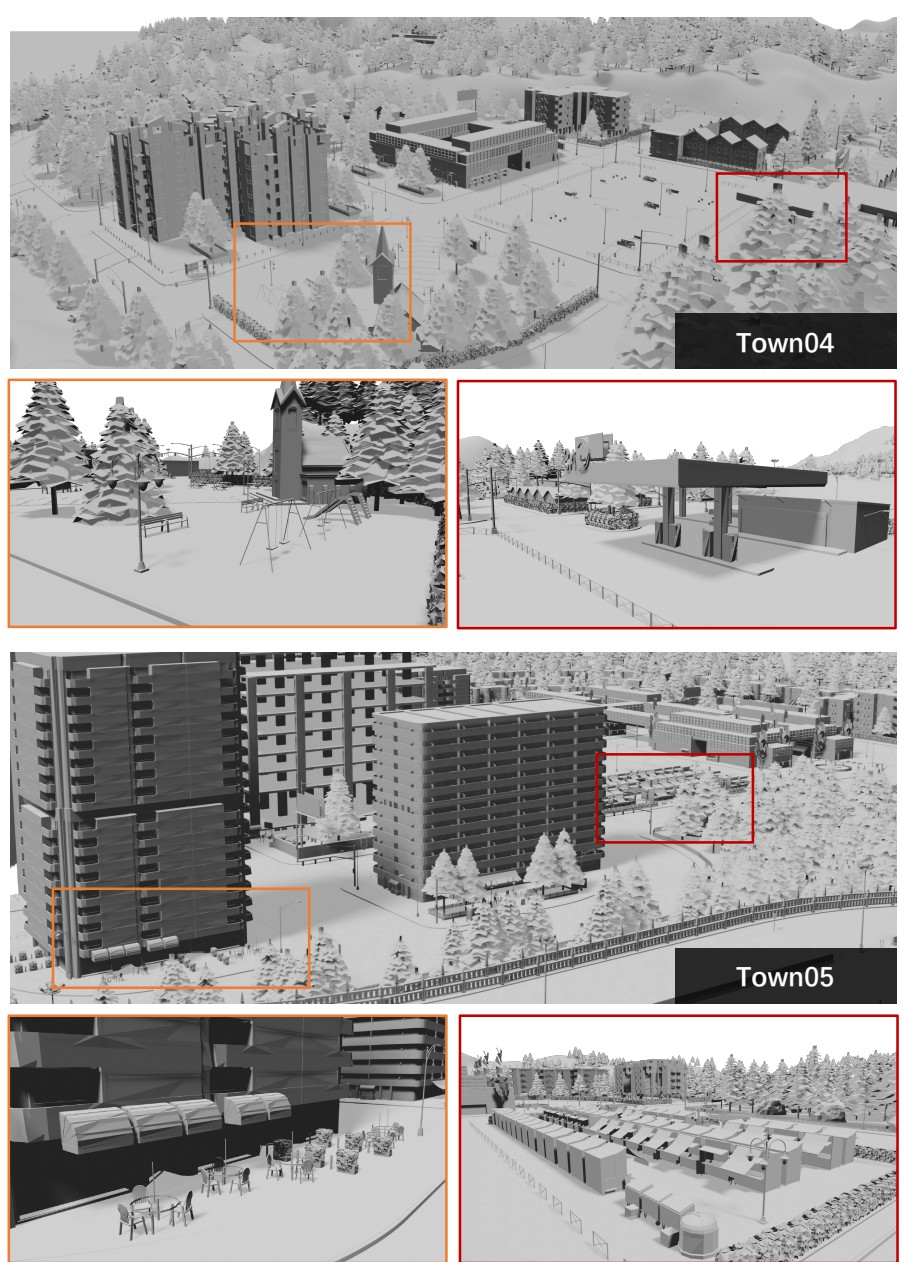

Figure 12: More visualizations of the complex and precise geometric structures included in the ground truth mesh of SS3DM.

