# OpenReview forum: "SS3DM: Benchmarking Street-View Surface Reconstruction with a Synthetic 3D Mesh Dataset"
_NeurIPS.cc/2024/Datasets_and_Benchmarks_Track — NeurIPS 2024 Track Datasets and Benchmarks Poster_

### Official Review · Reviewer_3RDx · 2024-07-12
**A synthetic dataset based on CARLA for surface reconstruction in street view.**

**Rating:** 5
**Confidence:** 4
**Correctness:** Yes
**Clarity:** Yes

**Review:**

### Quality & Clarity
The paper presents a detailed introduction to the SS3DM dataset and showcases its utility in benchmarking street-view surface reconstruction. The writing is clear. While the methods and metrics used for evaluation are robust and well-explained, the reliance on synthetic data raises concerns about its applicability to real-world scenarios.

### Originality & Significance
The work is original in its approach to creating a synthetic dataset specifically tailored for street-view surface reconstruction. However, the heavy dependence on synthetic data generated from the CARLA simulator detracts from its originality, as it does not fully address the complexities and variabilities of real-world environments.

**Strengths:**

**Comprehensive Benchmarking**: Includes extensive evaluation metrics and analysis of current state-of-the-art methods for surface reconstruction and novel view synthesis.

**Additional Feedback:**

The reliance on synthetic data from the CARLA simulator raises concerns about the dataset's applicability and relevance to real-world street-view surface reconstruction tasks. Synthetic environments often lack the unpredictability and variability of real-world scenarios, which are crucial for robust evaluation.

**Documentation:**

Yes

**Limitations:**

Yes

**Opportunities For Improvement:**

**Synthetic Nature**: The reliance on synthetic data from the CARLA simulator raises concerns about the dataset's applicability and relevance to real-world street-view surface reconstruction tasks. Synthetic environments often lack the unpredictability and variability of real-world scenarios, which are crucial for robust evaluation.

**Relation To Prior Work:**

Yes

**Summary And Contributions:**

The paper introduces SS3DM, a synthetic dataset designed to benchmark street-view surface reconstruction using precise 3D mesh models created with the CARLA simulator. Unlike traditional datasets like KITTI and Waymo, which provide noisy LiDAR points, SS3DM offers high-fidelity ground-truth meshes, allowing for accurate geometric and surface normal assessments. The dataset includes multi-view RGB video sequences and LiDAR points, collected by simulating driving scenarios with advanced sensors. Key contributions include the development of a plugin for exporting detailed 3D meshes, extensive benchmarking of current surface reconstruction methods, and the identification of challenges and future research directions in this field.

---

> ### Author Rebuttal · Authors · 2024-08-17
>
> Thank you for the insightful and constructive comments. We genuinely appreciate your recognition of the originality of our work, extensive evaluation metrics, and analysis.
>
> **Synthetic Nature**:
>
> (1) About applicability to real-world street-view surface reconstruction task: Thank you for pointing out the challenges in addressing domain gaps between synthetic and real-world data. Following reviewer 1uab’s suggestion, we will manually construct and annotate partial street-view mesh models from real-world datasets like Waymo Block-NeRF and make use of existing sim2real techniques and multi-source domain adaptation methods to conduct some analyses in future works.
>
> (2) About lacking the unpredictability and variability of real-world scenarios: Thank you for pointing out this important issue. Following reviewer Mh8j’ suggestion, we will extend SS3DM to include dynamic objects, traffics, and holistic reconstructions (i.e., both dynamic and static reconstruction) as another benchmark.
>
> For details about (1) and (2), please refer to the general response to all reviewers.

---

### Official Review · Reviewer_Mh8j · 2024-07-23

**Rating:** 7
**Confidence:** 3
**Correctness:** Yes
**Clarity:** Yes

**Review:**

See strengths and weakness.

**Strengths:**

- SS3DM provides precise synthetic 3D mesh models as ground-truth geometry, enabling accurate evaluation of reconstructed surface positions and normals. This is an improvement over noisy LiDAR points provided in real-world datasets like KITTI and Waymo.
- SS3DM Contains multi-view RGB video sequences and multi-view LiDAR point clouds captured from a realistic virtual car sensor setup, mimicking advanced autonomous driving systems.
- SS3DM Covers diverse street scenes exhibiting varied structures like buildings, overpasses, yards, fences and poles.
- It makes the dataset, CARLA exportation plugin, and benchmark code publicly available to facilitate future research.

**Additional Feedback:**

N/A

**Documentation:**

Yes

**Limitations:**

Yes

**Opportunities For Improvement:**

- As a synthetic dataset, SS3DM likely suffers from domain gaps compared to real-world street-view data. Models trained on SS3DM may not generalize as well to real scenarios without additional domain adaptation techniques.
- The diversity of scenes is still somewhat limited compared to the full variability of real-world environments. SS3DM currently lacks dynamic objects like moving vehicles and pedestrians.

Overall, I understand it's extremely challenging to collect a similar dataset in the real world. So I still thinks this is a good paper and should be presented at NeurIPS.

**Relation To Prior Work:**

Yes

**Summary And Contributions:**

The paper introduces SS3DM, a new synthetic dataset specifically designed for benchmarking surface reconstruction methods in street-view outdoor scenes. The key features and contributions of SS3DM include: It provides precise synthetic 3D mesh models as ground-truth geometry for large-scale street scenes. This enables accurate evaluation of reconstructed surface positions and normals, going beyond the noisy LiDAR points provided in existing street-view datasets like KITTI and Waymo. Using SS3DM, the authors conduct an extensive benchmark of state-of-the-art surface reconstruction methods. They analyze the methods' limitations and discuss key challenges in street-view surface reconstruction. The authors make the dataset, CARLA exportation plugin, and benchmark code publicly available to facilitate future research in this area.

---

> ### Author Rebuttal · Authors · 2024-08-17
>
> Thank you for the valuable and encouraging comments. We strongly agree with the comment that “I understand it's extremely challenging to collect a similar dataset in the real world”, and we really appreciate the comment “So I still thinks this is a good paper and should be presented at NeurIPS.”
>
> 1. **Domain Gap and Sim2Real Generalization**: Thank you for pointing out the challenges in overcoming domain gaps and for the valuable suggestions about domain adaption techniques. Please refer to the general response to all reviewers.
>
> 2. **Diversity of scenes**: We really appreciate your suggestions to improve the dataset diversity by including dynamic objects. Following your suggestion, we have started to extend SS3DM during the rebuttal period by including dynamic objects and traffics utilizing the CARLA traffic functionalities. Please refer to Figures R1 and R2 in the figure PDF submitted for rebuttal; we could add moving vehicles/pedestrians in the street scenes and extract the ground truth meshes for dynamic objects at every timestamp. We will include more examples of dynamic objects in our revised paper. Additionally, we plan to include holistic reconstruction (i.e., both dynamic and static reconstruction) as another benchmark in future work, which will be promptly updated on the maintained, publicly available dataset website.

---

### Official Review · Reviewer_1uab · 2024-07-26
**3D mesh reconstruction for street-view data benchmark**

**Rating:** 7
**Confidence:** 4

**Review:**

Pros:
- Clarity: The paper was well-written.
- Originality: The benchmark and the dataset are novel
- Significance: This work made clear contributions and would be of interest to the broad community including autonomous driving, robotics, etc.

Cons:
- Clarity: Some information needs to be clearly explained (please see Clarity section)

**Strengths:**

- The benchmark is of interest to the 3D research community as reconstructing precise out-door 3D scenes is a challenging task. The dataset provides clean ground-truth triangle meshes for the scenes, which have not existed before in prior works.
- The dataset was well-designed, including various modalities that can serve as inputs (RGB streams and LiDARs), with ground-truth camera parameters. By making use of synthetic data in simulation, the dataset parameters can be precisely controlled, which is promising for future research and analyses.
- The data covers 3 levels of difficulty: short, medium and long sequences. This is helpful for diagnosing and understanding the performance of baselines.
- Experiments were done thoroughly on various SOTA 3D reconstruction models that take in different input modalities. The findings show evidence that 3D mesh reconstruction of Street-view data is a challenging task that is worth further investigation.

**Additional Feedback:**

- Please explain the training/testing data splits and more details about the baseline methods
- Main table 3 can be present better that can clearly show the performance of models with different input modalities

**Clarity:**

- In general, the paper is well-written. The motivation and potential applications of the task were explained carefully.
- Table 3 can be present better. I don't think it's fair to compare methods that take in different modalities with each other. I would suggest splitting into 3 groups depending on the input modalities: RGB images, LiDAR and both.

**Correctness:**

Claims made in the paper are correct. The dataset was constructed carefully, providing annotations to a wide range of modalities. Experiment designs and evaluation methods were performed correctly and aligned with the goal of the paper.

**Documentation:**

Data license was mentioned. The authors promised to release the data generation pipeline. There is no more information regarding other details such as maintenance and availability plans.

**Ethics:**

No ethical concern

**Limitations:**

The paper discussed the limitations in detail and highlighted interesting directions for future research.

**Opportunities For Improvement:**

- For synthetic data there is always a concern about sim2real generalization. Can these baselines be trained on the proposed dataset and evaluated on real-world data like Waymo Block-NeRF or MatrixCity? Since there is no ground-truth 3D geometric information in these datasets, quality of the output can be converted to the format of the outputs of these datasets and evaluated on their established metrics. It is worth understanding the domain gap between this clean synthetic data with real-world scenarios.
- Details about data splits (train/val/test) were missing. How much data were the models trained on? Was there any pre-training involved?
- One of the standard metrics for 3D reconstruction task is IoU between the predicted and the ground-truth meshes. While this metric is not perfect, it is different from both distance-based metrics and surface normals. Was there a particular reason why this metric was not used?

**Relation To Prior Work:**

The paper discussed prior work clearly and demonstrated important contributions regarding the dataset and the proposed benchmkar.

**Summary And Contributions:**

- The paper introduced 3D mesh reconstruction benchmark for Street-view data, SS3DM, accompanied by the synthetic street-view dataset extracted from CARLA simulator. This dataset provides annotated camera poses, LiDARs, RGB streams, ground-truth triangle meshes in addition to depth maps and semantic information.
- Multiple SOTA 3D reconstruction methods were evaluated on SS3DM benchmark using different input modalities.

---

> ### Author Rebuttal · Authors · 2024-08-17
>
> Thank you for the valuable and encouraging comments.
>
> 1. **Understand the domain gap with real-world datasets**: We really appreciate the suggestion to understand the domain gap by utilizing real-world datasets like Waymo Block-NeRF, or MatrixCity. Additionally, we agree that the lack of geometric ground truth in real-world dataset is an extremely challenging problem. We plan to manually construct and annotate partial street-view mesh models in real-world datasets like Waymo Block-NeRF, and utilize them to develop sim2real methods in our future work. It would be interesting to learn from previous sim2real techniques [A, B] and develop our methods on the street-view surface reconstruction task. We will mention this point as future work in our revised paper, and promptly update the sim2real generation evaluation on the maintained dataset website.
>
>
> 2. **Details about data split**: Thank you for the comment. Our SS3DM dataset can be used in two ways: (1) purely as a test data and (2) for training and validating by splitting into train/val/test splits. In our experiments, we follow the first way, i.e., treating all the data sequences as test data and involve no pre-training. We will make this point clear in the revision.
>
>
> 3. **More details about the baseline methods**: Thank you for the comment. Some details about the baseline methods have been presented in Section A.2 of Appendix. We will make the dataset and the code publicly available, with comprehensive details provided on a website that will be regularly maintained and updated.
>
>
> 4. **Evaluation of IoU Metric**: Thank you for suggesting the IoU metric. We evaluated the IoU metric based on 0.05m voxelization and reported the results in the first column of Table R1 in the figure PDF submitted for rebuttal. We’ll include the IoU results in our revised paper.
>
> 5. **Maintenance and availability plans**: Thank you for this comment and suggestion. Please refer to the general response to all reviewers.
>
> 6. **Present Table 3 better**: Thank you for the suggestion. We have reorganized the methods to group those using the same modality together. Please refer to Table R1 in the figure PDF submitted for rebuttal. We’ll update Table 3 in our revised version accordingly.
>
> [A] Hu, Xuemin, et al. "How simulation helps autonomous driving: A survey of sim2real, digital twins, and parallel intelligence." IEEE Transactions on Intelligent Vehicles (2023).
>
> [B] Zhao, Sicheng, et al. "Multi-source domain adaptation in the deep learning era: A systematic survey." arXiv preprint arXiv:2002.12169 (2020).

---

> > ### Comment · Reviewer_1uab · 2024-08-31
> >
> > I appreciate the authors for their effort in addressing my comments. I will keep the score.

---

### Author Rebuttal · Authors · 2024-08-17

# General Response to All Reviewers:


We thank the three reviewers for the constructive comments, which help us improve the paper. We appreciate all the reviewers for the recognition of our novel dataset construction and comprehensive benchmarking experiments. Especially, Reviewer 1uab points out that “This work made clear contributions and would be of interest to the broad community including autonomous driving, robotics, etc.,” Reviewer Mh8j points out that “Overall, I understand it's extremely challenging to collect a similar dataset in the real world. So I still thinks this is a good paper and should be presented at NeurIPS,” and Reviewer 3RDx comments that “The work is original in its approach to creating a synthetic dataset specifically tailored for street-view surface reconstruction.” Following the reviewers’ suggestions and addressing the common sim2real generalization problem, we’d like to clarify the following points. All the responses in rebuttals will be put into the revised manuscript.

1.	**Publicity and maintenance of the constructed SS3DM dataset**. To address the reviewer 1uab’s documentation comment, we will follow the good practices from the Waymo dataset and maintain a publicly available website for our SS3DM dataset. We will publicly release the full dataset and all the codes for benchmarking and data exporting. We will also keep the dataset updated by regularly responsing to community feedback and adding in newly requested features. In particular, we plan to ensure long-term accessibility by hosting the dataset on a stable platform with version control to solve any reported issues, providing periodic data and metric updates, and addressing discussions in the community.
2.	**Interesting study on sim2real generalization**. We agree with the reviewers that sim2real generalization is challenging and we appreciate the reviewers’ valuable advice on mitigating the domain gap. Following reviewers 1uab and Mh8j’s suggestions, we would manually construct and annotate street-view mesh models for partially data in the real-world datasets like Waymo Block-NeRF, and utilize them to develop sim2real domain adaptation methods. It would be interesting to learn from previous sim2real techniques [A] and multi-source domain adaptation methods [B] to develop suitable methods for the sim2real generalization on the street-view surface reconstruction task. We will mention this point as future work in the revision. In addition, the newly developed evaluations for the sim2real generalization will be promptly updated on the maintained dataset website.
3.	**Extending SS3DM to include dynamic objects**. We thank reviewer Mh8j for the suggestion. We started to extend SS3DM by including dynamic objects and traffics during the rebuttal period. In the figure PDF submitted to rebuttal, we present two typical examples of dynamic objects in SS3DM; i.e., Figure R1 depicts an example of RGB images with dynamic objects, and Figure R2 illustrates an example of ground truth mesh for the dynamic objects. We will include these and more examples of dynamic objects in our revised paper. Meanwhile, we plan to include holistic reconstruction (i.e., both dynamic and static reconstruction) as another benchmark in future work, which will be promptly updated on the maintained dataset website.


Once again, we thank the three reviewers for their valuable comments and suggestions.

[A] Hu, Xuemin, et al. "How simulation helps autonomous driving: A survey of sim2real, digital twins, and parallel intelligence." IEEE Transactions on Intelligent Vehicles (2023).

[B] Zhao, Sicheng, et al. "Multi-source domain adaptation in the deep learning era: A systematic survey." arXiv preprint arXiv:2002.12169 (2020).

---

### Decision · Program_Chairs · 2024-09-26

**Decision:**

Accept (Poster)

**Comment:**

This paper has somewhat mixed reviews (7,7,5). Two reviewers appreciated the value and novelty of this benchmark for the challenging scenario of reconstructing outdoor scenes in 3D. One reviewer believes that, due to building on the CARLA simulator, the work offers only minor originality and also lacks the realism of real scene. On balance, the Area Chair believes that, despite these valid points, the paper offers sufficiently strong material for acceptance, and in particular believes the benchmark will be adopted as it does cover a hole in the current landscape of datasets.